# Redox Mechanisms of Platelet Activation in Aging

**DOI:** 10.3390/antiox11050995

**Published:** 2022-05-19

**Authors:** Sean X. Gu, Sanjana Dayal

**Affiliations:** 1Department of Laboratory Medicine, Yale School of Medicine, New Haven, CT 06511, USA; sean.gu@yale.edu; 2Department of Internal Medicine, University of Iowa Carver College of Medicine, Iowa City, IA 52242, USA; 3Iowa City VA Healthcare System, Iowa City, IA 52246, USA

**Keywords:** aging, oxidative stress, vascular disease, platelets

## Abstract

Aging is intrinsically linked with physiologic decline and is a major risk factor for a broad range of diseases. The deleterious effects of advancing age on the vascular system are evidenced by the high incidence and prevalence of cardiovascular disease in the elderly. Reactive oxygen species are critical mediators of normal vascular physiology and have been shown to gradually increase in the vasculature with age. There is a growing appreciation for the complexity of oxidant and antioxidant systems at the cellular and molecular levels, and accumulating evidence indicates a causal association between oxidative stress and age-related vascular disease. Herein, we review the current understanding of mechanistic links between oxidative stress and thrombotic vascular disease and the changes that occur with aging. While several vascular cells are key contributors, we focus on oxidative changes that occur in platelets and their mediation in disease progression. Additionally, we discuss the impact of comorbid conditions (i.e., diabetes, atherosclerosis, obesity, cancer, etc.) that have been associated with platelet redox dysregulation and vascular disease pathogenesis. As we continue to unravel the fundamental redox mechanisms of the vascular system, we will be able to develop more targeted therapeutic strategies for the prevention and management of age-associated vascular disease.

## 1. Introduction

Aging is a biological phenomenon in living organisms that is characterized by a gradual decline in physical and mental capacity [1]. It represents an accumulation of adverse changes over time that increases the risk of disease and, ultimately, death. Cardiovascular disease is the leading cause of death in the United States [2] and worldwide [3], and it is well established that myocardial infarction and stroke increase significantly with age [4]. These observations are, in part, due to the high prevalence of associated comorbid conditions (e.g., diabetes, obesity, hypertension, hyperlipidemia, etc.) that are frequently observed in the elderly [5]. As the average human life expectancy and, consequently, the number of elderly in the population are expected to grow in the coming decades, the burden of chronic diseases, including cardiovascular and cerebrovascular disease, is projected to have a more significant impact on human health and healthcare costs [1,6,7]. While several cell types are known to contribute to vascular pathologies, this review will focus on platelets as the key mediators in the progression of thrombotic vascular disease (Figure 1).

## 2. Platelets and Thrombotic Vascular Disease

Platelets are anucleate cells derived from megakaryocytes that are key components of the vascular system, and they are traditionally recognized for their vital role in hemostasis through the enhancement of coagulation and thrombus formation at sites of vascular injury [8]. Platelet activation is central to thrombus formation and, thus, plays a critical role in the pathogenesis of thrombotic vascular diseases [9,10,11]. In addition to their fundamental roles in hemostasis and thrombosis, platelets can also act as mediators of inflammation and immunity through functional interactions with the vascular endothelium and circulating hematopoietic cells [10,12,13,14]. Altered platelet function and platelet hyperactivity have been associated with aging. Early epidemiological and functional studies suggested that platelet activity is enhanced in elderly individuals, although the precise mechanisms for these changes were not entirely clear [15,16,17,18]. Below, we discuss the recent literature on the redox mechanisms that regulate platelet activation while highlighting specific alterations that occur with age. In addition, we will discuss the role of platelets in cardiovascular comorbidities that are more prevalent in the elderly and can provide mechanistic insights into age-associated vascular disease. Understanding the fundamental cellular and molecular processes that occur in platelets with aging will provide opportunities to develop novel therapeutic strategies to prevent age-associated vascular pathologies and reduce the burden of disease in the elderly.

## 3. Oxidative and Antioxidative Mechanisms in Regulation of Platelets in Aging

It is well recognized that there is an age-associated increase in reactive oxygen species (ROS), and oxidative modifications to macromolecules (i.e., DNA, proteins, lipids) have been implicated as fundamental drivers of age-associated pathologies [19]. In the vascular system, accumulating evidence has demonstrated critical roles for ROS in the regulation of a variety of cellular and molecular processes, which, when dysregulated, can increase disease burden.

The relationship between oxidative stress and platelet activity has been a growing area of interest in studying the mechanisms of cardiovascular diseases and other age-related thrombotic vascular conditions. Initial studies in the 1970s first demonstrated the capacity of platelets to generate superoxide (O_2_^●−^) [20], a highly reactive species with a relatively short half-life. Several studies subsequently have demonstrated that exogenous and endogenous O_2_^●−^ enhance platelet function, which can be reversed in the presence of SOD enzymes or O_2_^●−^ scavengers [21,22]. Superoxide can be generated by several sources in the vasculature, including nicotinamide adenine dinucleotide phosphate (NADPH) oxidase, uncoupled endothelial nitric oxide synthase (eNOS), xanthine oxidase, and mitochondrial sources [23]. Studies in our lab found that platelets isolated from aged mice displayed enhanced activation responses compared to those from young mice, which was prevented by the NADPH oxidase inhibitor apocynin [24]. Platelets from aged mice also had increased mRNA expression of the p47^phox^ NADPH oxidase subunit. However, the mRNA levels for another subunit, Nox2, were not significantly different between platelets from aged and young mice, and the mRNA for Nox1 and Nox4 was not detectable in these studies [24]. The Nox family consists of seven isoforms, of which four (Nox1, 2, 4, and 5) are found in vascular cells [25]. Nox proteins are the catalytic component of the NADPH oxidase and, together with p22^phox^, comprise the large heterodimeric cytochrome subunit [26]. The Nox2 NADPH oxidase isoform consists of additional cytosolic regulatory subunits that include p47^phox^, p40^phox^, and p67^phox^. Further, studies in our lab found that Nox2 is not an essential source of platelet ROS and is not significantly involved in platelet activation and arterial thrombosis in young mice [27]. Some studies have reported a limited role of Nox2 in GPVI-dependent platelet responses [28], while other studies have found that Nox2 is important for collagen and thrombin-induced platelet responses in mice [29]. Using an EPR-based technique to measure intracellular platelet ROS, Vara et al. further elucidated the differential roles of Nox1 and Nox2 in platelet activation [30]. They found that Nox1 is important for collagen-induced platelet responses and that intracellular but not extracellular O_2_^●−^ was critical for platelet activation by collagen, while Nox2 is important for thrombin-induced platelet activation [30]. Subsequent studies by the same group reported that a combined triple deficiency of Nox1, Nox2, and Nox4 in mice resulted in impaired platelet aggregation responses and decreased susceptibility to experimental thrombosis, while knockouts of single Nox isoforms showed no significant vascular effects in mice [31]. These studies indicate that Nox proteins are important for platelet activation and thrombosis in vivo, but there is redundancy in the O_2_^●−^ generating system, given that single knockouts of individual Nox proteins produced minimal functional effects on platelets.

Studies have demonstrated that hydrogen peroxide (H_2_O_2_) can also be released from platelets [32]. H_2_O_2_ has a relatively long half-life and is an important intracellular signaling molecule, given its ability to diffuse across cellular membranes [33]. The conversion of O_2_^●−^ to H_2_O_2_ is catalyzed by superoxide dismutase (SOD), which consists of three major isoforms in mammals, copper-zinc SOD (SOD1), manganese SOD (SOD2), and extracellular SOD (SOD3), that differ in their cellular localization and metal cofactors in the catalytic site [34]. The major antioxidant enzymes responsible for the catalytic inactivation of H_2_O_2_ include catalase, glutathione peroxidase (GPx), thioredoxin (Trx), and peroxiredoxin (Prdx). In humans, decreased activity of GPx-1 has been associated with both platelet hyperreactivity and an increased risk of cardiovascular events [35,36,37,38]. Studies in humans have also demonstrated that platelet activation is associated with the production of H_2_O_2_, and pre-treatment with catalase eliminated platelet H_2_O_2_ and inhibited collagen-induced platelet aggregation ex vivo [39]. Studies from our lab demonstrated that aged mice also exhibited increased susceptibility to thrombosis under experimental conditions and were protected from this phenotype by transgenic overexpression of intracellular GPx-1 [24]. Furthermore, platelets from aged mice displayed increased expression of not only the p47^phox^ NADPH subunit but also SOD-1, providing a mechanistic explanation that age-dependent platelet hyperactivation is mediated by increased platelet O_2_^●−^ generation and its conversion to H_2_O_2_ intracellularly [24]. These findings are in concordance with a more recent study using both human and mouse platelets that found that thrombin-induced platelet responses are dependent on the dismutation of O_2_^●−^ to H_2_O_2_ [30]. In a separate study, Jin et al. demonstrated enhanced platelet aggregation responses and increased susceptibility to pulmonary thromboembolism and thrombotic stroke in mice with a deficiency of the circulating H_2_O_2_ metabolizing enzyme GPx-3 [40]. Together, these studies indicate a protective role for endogenous GPx in age-associated thrombotic consequences.

In a recent elegant study, Jain et al. stratified high cardiovascular risk patients into several age cohorts and measured the redox changes in platelets compared to younger healthy individuals [41]. A progressive age-associated increase in platelet reactivity and intracellular ROS was observed up until ~80 years of age, whereby a decline in platelet reactivity and ROS was seen. These findings were attributed to a decline in platelet antioxidants, including SOD-1, GPx-1, Prdx-6, and catalase, in patients aged 60–79 years and then an upregulation of these enzymes in people aged 80 or older that likely lowered the intracellular ROS burden. Using cross-sectional and longitudinal aging studies in mice, the authors recapitulated the findings of the human cohort [41]. These findings suggest that in the very elderly (i.e., 80 years or older), platelet antioxidant responses may be elicited as an adaptive mechanism to counterbalance the increase in oxidative stress and platelet hyperactivation associated with aging [42,43].

## 4. Platelet Mitochondria in Aging

Another important cellular source of ROS is mitochondria, which are key sites of the tricarboxylic acid (TCA) cycle and oxidative phosphorylation. Oxygen consumption in the mitochondria is central to these processes. Its utilization to generate ATP through the electron transport chain is not completely efficient, and if electron transfers occur prematurely to oxygen, O_2_^●−^ and its subsequent conversion to H_2_O_2_ can be generated as by-products [44]. Initially proposed by Denham Harman in the 1950s, the free radical theory of aging aimed to link aging with oxidative stress [45]. Subsequent revisions to this theory have focused on mitochondria as the primary source of ROS, and they are hypothesized to promote aging and age-related diseases through the accumulation of oxidative damage to protein, DNA, and lipid [46]. Although some studies have supported this theory [47,48], other observations have questioned a direct causal relationship between ROS and aging [49,50]. In fact, some studies have demonstrated increased lifespans in various model organisms with elevated mitochondrial ROS [51,52,53]. Thus, there is some controversy in the literature on the precise role of mitochondrial ROS in aging, which may reflect the different methods and types of model organisms used in the separate studies.

Despite not having a nucleus, platelets contain functional mitochondria and are metabolically active [54]. Platelets consume high levels of ATP, and mitochondrial aerobic respiration provides approximately 40% of the total basal energy requirements, while the remaining 60% is generated through glycolysis [54]. Studies have demonstrated that platelet activation is associated with alterations in mitochondrial processes such as the generation of ROS, induction of mitochondrial permeability transition pore (MPTP) formation, loss of mitochondrial membrane potential (ΔΨ_m_), and induction of apoptosis [55,56,57,58]. Conversely, inhibition of mitochondrial respiratory function inhibits platelet aggregation and reduces clot formation [59]. A recent study provided evidence that platelet mitochondrial function is altered with age [60]. It was observed that older individuals (88 ± 2 years) display significant alterations in platelet bioenergetics, such as lower basal and ATP-linked respiration, compared to younger individuals (26 ± 5 years). Furthermore, an increased proton leak was observed in the platelet mitochondria of older individuals; this was attributed to the upregulation of an uncoupling protein (UCP). Previous studies have shown that UCPs are upregulated by superoxide and protected from ROS production through a decreased flow of electrons through the electron transport chain [61]. This process likely serves as an adaptive response to increased levels of mitochondrial ROS with advancing age in order to dissipate ROS generation [62,63]. Therefore, future studies could focus on the protective and harmful effects of key mitochondrial proteins regulating ROS within platelets during aging.

Mitochondrial superoxide overproduction has been shown to potentiate agonist-induced platelet activation under hyperglycemic conditions [64,65,66]. Moreover, hyperglycemia induces mitochondrial dysfunction and superoxide production in platelets that can induce both platelet hyperactivation and apoptosis, leading to increased susceptibility to experimental thrombosis in murine models of diabetes [66]. Despite observations that SOD2, the mitochondrial-specific antioxidant enzyme that dismutates O_2_^●−^ to H_2_O_2_, is upregulated in patients with type 2 diabetes mellitus [67], studies failed to show significant alterations in platelet function or thrombotic susceptibility with platelet-specific knockouts of SOD2 in non-aged non-diabetic murine models [68]. Interestingly, these mice displayed increased mitochondrial superoxide production, while total intracellular ROS was unchanged compared to wild-type mice [68]. It is possible that the relatively low basal production of mitochondrial superoxide within platelets in young mice is insufficient to induce pathologic vascular changes, even with a concomitant deficiency of SOD2. Given the evidence that platelet activation and thrombotic risk are increased with age, it will be important for future studies using aging models to evaluate the effects of mitochondrial ROS and antioxidant enzymes in age-associated vascular disease and thrombosis.

Beyond its role in energy and redox metabolism, there is growing appreciation for mitochondria in other processes, including inflammation, stress response, and cell death, that can impact longevity and contribute to age-associated vascular diseases [69,70,71,72,73,74]. Autophagy is a fundamental cellular process that functions to degrade cellular contents and limit the accumulation of damaged biomacromolecules and organelles [75,76]. It helps in maintaining homeostasis during cellular stress and is presumed to prevent physiologic aging [77]. Autophagy has been demonstrated in platelets and has been shown to be important for platelet function and can impact hemostasis and thrombosis when dysregulated [72]. A pathway downstream to ROS in aging could be a mechanistic target of rapamycin complex 1 (mTORC1), which is involved in nutrient homeostasis and is closely integrated with the autophagy machinery [78]. A study showed that mTORC1 is upregulated in aged mice in a ROS-dependent pathway, and pharmacologic or genetic silencing of mTORC1 within platelets reduced the susceptibility to venous thrombosis in murine models [71].

Mitophagy is an autophagic process that is selective for removing damaged and dysfunctional mitochondria [79,80]. Alterations in mitophagy can impact platelet life span through direct interactions of apoptosis proteins with the mitophagic machinery [81]. Studies have demonstrated that platelets of diabetic patients are susceptible to oxidative stress, which can induce the phosphorylation of p53, resulting in mitochondrial dysfunction and apoptosis that can contribute to vascular thrombosis [66,82]. In parallel, increased ROS can induce selective mitophagy in human platelets, and this process serves as a protective mechanism against oxidative stress to remove damaged mitochondria and prevent apoptosis in the platelets of patients with diabetes [82]. In the same study, disruption of mitophagy using mice with a genetic deletion of Parkin or PINK1 produced platelets with increased susceptibility to H_2_O_2_-induced mitochondrial damage and apoptosis [82]. Moreover, genetic deletion of PINK1 in a diabetic mouse model produced platelets that were hyperactive with increased P-selectin surface expression, and the mice exhibited increased susceptibility to experimental carotid artery thrombosis [82].

Interestingly, platelet mitophagy can be regulated by methionine oxidation (MetO) [83], which is a reversible post-translational modification on proteins implicated in both aging and vascular disease [84,85,86,87,88]. In diabetic patients, increased oxidative stress was found to increase MetO-modified proteins. Specifically, Parkin Met192 can be oxidized and lead to protein aggregation and the disruption of mitophagy in human platelets [83]. In the same study, genetic ablation of the mitochondrial-specific methionine sulfoxide reductase MsrB2 also disrupted mitophagy and promoted the apoptosis of murine platelets [83]. Additional evidence indicates that MsrB2 is released from damaged mitochondria, reduces/reverses Parkin Met192 oxidation through direct interactions, and initiates mitophagy to prevent platelet apoptosis [83]. These findings represent a novel redox mechanism to regulate platelet mitophagy and apoptosis under conditions of increased oxidative stress, such as diabetes.

## 5. Platelets and Inflammation in Aging

The term “inflammaging” was a concept introduced in 2000 by Dr. Franceschi and is now commonly used to describe the pathologic consequences of chronic low-grade inflammation and physiologic stimulation of the innate immune system that occurs with advancing age [89]. This concept has been used to explain the higher prevalence of chronic disorders in the elderly, including obesity, type 2 diabetes mellitus, and cardiovascular diseases [89]. Platelets are also recognized as immune and inflammatory cells [13,90]. The immune functions of platelets are evidenced by their direct interactions with vascular and inflammatory cells and by the presence of various cytokines and chemokines contained in their granules and cytoplasm [13,90].

A recent article by Davizon-Castillo et al. provides compelling evidence of a link between inflammation and platelet activity [91], which is also discussed in several accompanying commentaries [92,93]. TNF-α is a key inflammatory cytokine that is highly correlated with age-associated cardiovascular disease and is intricately linked with ROS [94,95,96,97]. It was observed that both aged humans and mice exhibited elevated plasma TNF-α and platelet hyperactivation [91]. Utilizing several complementary murine models of TNF-α elevation or depletion, a direct functional effect of TNF-α in platelet hyperactivation during aging was demonstrated [91]. Single-cell transcriptome analysis of the bone marrow compartment showed significant reprogramming in platelet/megakaryocyte progenitor populations in aged mice that corresponded to alterations in mitochondrial function, oxidative phosphorylation, and inflammatory signaling pathways [91]. These findings are largely consistent with a prior study that also showed changes in platelet/megakaryocyte progenitors using murine models of aging [98]. In addition, Davizon-Castillo et al. observed increased mitochondrial mass and altered bioenergetics with increased oxygen consumption in platelets from aged mice that were mitigated with TNF-α blockage [91], which provides further evidence for an intriguing role of mitochondria in “inflammaging”. Phagosome maturation was one of the top pathways identified by Ingenuity Pathway Analysis software [91], and, as discussed above, autophagy/mitophagy is important for maintaining platelet function in the presence of Oxidative stress and mitochondrial damage [82]. Overall, Davizon-Castillo et al. provide substantial evidence that TNF-α is a crucial driver of platelet hyperreactivity during aging. Their findings provide a good rationale for future aging studies to examine the precise mechanisms of how inflammatory, mitochondrial, and oxidative pathways may converge to induce aberrant platelet hyperactivity and their pathological impact on age-associated thrombotic diseases.

In other studies, stimulation of TLRs on platelets has also been shown to modulate TNF-α production in vivo, likely through interactions with other immune or vascular cells [99]. Platelets express several toll-like receptor (TLR) family members [99,100] that have classical functions not only in mediating innate immune signaling but also in regulating platelet function [101,102,103,104]. Stimulation of TLR4 by lipopolysaccharide (LPS) enhances platelet activation and can promote direct interactions with inflammatory and immune cells such as neutrophils [101,105]. These effects are likely ROS-dependent, based on the observations that platelet ROS is increased with LPS stimulation, whereas treatment with antioxidant enzymes such as SOD or catalase prevents LPS-induced platelet activation [105]. Consistent with these findings, other studies have demonstrated that platelet TLR2 stimulation can also promote platelet activation and is associated with increased ROS production and platelet–neutrophil aggregates [106]. Given that LPS is a key outer membrane component of Gram-negative bacteria known to elicit robust inflammatory and immune responses, these findings may explain the observations of the increased risk of vascular events, such as MI and stroke, following acute infections that occur more commonly in the elderly population [107].

## 6. Oxidative Stress and Platelets in Chronic Diseases

Hyperlipidemia and atherosclerosis are significantly associated with advancing age and are major risk factors for the development of thrombotic vascular disease [5,108]. Early clinical studies observed enhanced platelet reactivity in patients with hyperlipidemia [109,110,111]. It was discovered that oxidized forms of lipids, such as oxidized low-density lipoprotein (oxLDL), can promote platelet activation and thrombosis in murine models of atherosclerosis via direct binding to CD36 [112], a multiligand scavenger receptor highly expressed on platelets as well as a broad range of other vascular cell types [113]. Redox-sensitive signaling pathways directly downstream of the CD36 receptor were identified to mediate oxLDL-induced platelet activation and thrombosis [114,115]. Specifically, the binding of oxLDL to the CD36 receptor on platelets induces sustained generation of intracellular ROS and promotes platelet aggregation, which was prevented by the pharmacologic inhibition of Nox2 in human platelets or the genetic ablation of NOX2 in murine platelets [115,116]. Other groups have shown that both NOX1 and NOX2 isoforms contribute to oxLDL-induced platelet activation [30]. It was demonstrated that a redox-sensitive protein, extracellular signal-regulated kinase 5 (ERK5) of the MAPK family, is directly activated by intracellular ROS generated through oxLDL-CD36 signaling and contributes to platelet activation, aggregation, and thrombus formation in vivo [117]. Moreover, this mechanistic pathway involving CD36 and ERK5 also promotes caspase activation and phosphatidylserine externalization on platelets that increase its procoagulant activity and support fibrin formation in vivo under conditions of dyslipidemia [118]. The CD36 signaling pathway can also activate innate immune signaling cascades in platelets through TLR activation, contributing to platelet hyperreactivity in the setting of hyperlipidemia [103]. The clinical relevance of the CD36 signaling pathway in thrombotic vascular disease is evidenced by human genetic studies identifying polymorphisms in the *CD36* gene that are strongly associated with platelet surface CD36 expression and risk of acute myocardial infarction [119,120]. Platelet CD36 signaling has also been suggested to play a role in platelet hyperreactivity and thrombosis in other age-related conditions associated with increased cardiovascular risk, including diabetes mellitus [121].

Type 2 diabetes mellitus (T2DM) is a well-established cardiovascular risk factor and is significantly more prevalent in the elderly population [122]. Patients with T2DM exhibit platelet hyperreactivity, and several groups have provided evidence that hyperglycemia induces alterations in oxidative pathways in platelets. For instance, the expression of the P2Y_12_ receptor has been shown to be significantly increased in platelets from T2DM patients and is constitutively activated [123]. Stimulation of the P2Y_12_ receptor under high glucose conditions in animal models has been shown to induce pathways that increase platelet intracellular ROS, contributing to platelet hyperactivity and limiting certain antiplatelet therapies [123]. There is evidence that platelets from patients with T2DM with poor glycemic control express higher levels of Nox1 [124]. Hyperglycemia also can induce mitochondrial superoxide generation, potentiating collagen-induced platelet activation [64]. Urinary levels of platelet thromboxane (TX) metabolites are elevated in patients with T2DM [125], and it was demonstrated that aldose reductase is an important enzyme that mediates hyperglycemia and collagen-induced platelet activation and TX release through a pathway involving ROS generation [126]. Moreover, aldose reductase also contributes to mitochondrial dysfunction/damage and platelet apoptosis in the setting of hyperglycemia [66]. Other pathways of mitochondrial-ROS-driven platelet activation in diabetes have been discussed in the section on platelet mitochondria in aging (see above).

Cancer is another disease often associated with aging. The incidence of many cancers increases dramatically with age, and older individuals are at greater risk for the development of advanced disease [127]. Accumulating evidence has indicated that platelets and tumor cells can interact reciprocally through direct binding or through the secretion and uptake of cellular factors that can influence immune and vascular responses [128,129]. These interactions have been shown to alter the key pathological processes related to cancer tumorigenesis [130,131] and metastasis [132,133,134] and have also provided novel diagnostic tools for cancer detection [135,136]. Elevations in ROS have been detected in many subtypes of cancers, and redox dysregulation has been implicated in cellular signaling pathways associated with both cancer development and clearance [137,138]. Many of the growth factors and cytokines that can induce ROS generation and exert biological effects on cancer are abundant in platelets and can be secreted upon stimulation [90,139,140,141]. However, only a few studies have directly examined platelet-specific ROS in cancer and chemotherapeutics. A small study reported increased oxidative and nitrative modifications of platelet proteins from patients with breast cancer [142]. A separate study provided evidence that metabolites of tamoxifen can increase the production of ROS through the activation of NADPH oxidase and promote platelet aggregation [143]. Further studies elaborating on the specific platelet redox mechanisms in different cancer subtypes will better determine the role of platelets and ROS in cancer pathogenesis.

Smoking is significantly associated with increased risk and mortality from cardiovascular disease [144]. Several early clinical studies provided the initial evidence that smoking potentiates platelet aggregation responses [145,146]. Subsequent studies have shown that smoking can induce platelet thrombus formation ex vivo in patients with coronary artery disease who are taking aspirin [147]. Several mechanisms involving redox dysregulation and nitric oxide bioavailability have been implicated in smoking-induced platelet hyperactivation and thrombosis [148,149,150]. Interestingly, other studies have associated smoking with a paradoxical decrease in platelet activation and a reduced recurrence of cardiovascular events in patients who are taking oral P2Y12 inhibitors (i.e., clopidogrel) [151,152,153,154,155]. The precise mechanism of this protection is unclear but likely involves the effects of smoking on hepatic cytochrome P450 enzymes and the complex metabolism of clopidogrel [156,157]. Overall, the current evidence suggests that smoking increases platelet reactivity and thrombotic risk but enhances the efficacy of clopidogrel therapy after thrombotic events.

## 7. Platelets in COVID-19 Pathogenesis

Coronavirus disease 2019 (COVID-19) is caused by the infection of severe acute respiratory syndrome coronavirus 2 (SARS-CoV-2). SARS-CoV-2 primarily targets the lung epithelial cells to induce respiratory distress but can also cause a systemic inflammatory response and vascular thrombosis [158,159]. Age plays an important role in COVID-19 pathogenesis, with severe illness and mortality disproportionally affecting the elderly. This is compounded by the higher prevalence of other traditional cardiovascular risk factors with advancing age, including diabetes, obesity, and hypertension. Vascular thrombosis and thrombo-inflammatory complications are major causes of morbidity and mortality in COVID-19, and platelets have been implicated as a key contributor to disease pathogenesis [160]. SARS-CoV-2 mRNA can be detected in the platelets of subsets of COVID-19 patients, suggesting that platelets can uptake SARS-CoV-2 mRNA [161,162]. In a study evaluating 115 consecutive patients with COVID-19, platelets were shown to be hyperactive with increased adhesion, aggregation, degranulation, and extracellular vesicle release compared to healthy individuals, contributing to alterations in cytokine and growth factor release [161]. Other groups have also reported hyperactive platelets in COVID-19 and demonstrated that platelets can associate with monocytes through platelet surface P-selectin and αIIbβ3 binding, which induces tissue factor expression by monocytes [163]. In fact, platelets from COVID-19 patients exhibit increased interactions with multiple leukocyte subsets, including neutrophils, lymphocytes, and monocytes [162].

Interestingly, neutrophil extracellular traps (NETs), which are web-like chromatin structures containing DNA–histone complexes and antimicrobial proteins released upon neutrophil activation, are increased in the plasma of COVID-19 patients and correlated with disease severity in some studies [164,165,166]. In a small case series of COVID-19 patients who developed ST-elevated myocardial infarction (STEMI), increased incidence and density of NETs were observed in the coronary thrombi of COVID-19 patients undergoing primary coronary intervention [167]. NET formation can be induced by activated platelets [101,168,169,170], and there is evidence that platelets contribute to NET formation in COVID-19, with reports of NET-containing microthrombi associated with platelet deposition in COVID-19 autopsies [164]. These findings are supported by other studies demonstrating that NET formation can be triggered by platelet-rich plasma from COVID-19 patients [165]. There is evidence of redox dysregulation in COVID-19, with increased Nox2 activity detected by plasma metabolites [171]. Nox2 is able to regulate platelet activation and NET formation in the lung [172]. Another potential mechanism of redox dysregulation in COVID-19 is through altered iron homeostasis [173,174]. Studies early in the pandemic reported an association between increased serum ferritin and in-hospital mortality [175]. Iron overload is associated with ferroptosis, a form of regulated cell death characterized by the iron-dependent accumulation of lipid peroxides and redox dysregulation through the depletion of glutathione and the inhibition of GPx-4 [176,177]. Ferroptosis has been implicated in COVID-19 pathogenesis and is suggested to contribute to multiorgan damage [178]. The release of free iron from heme has been reported to increase lipid peroxidation and induce platelet activation and cell death through ferroptosis [179]. Nevertheless, the precise role of ROS and platelets in COVID-19 remains to be determined.

## 8. Conclusions

Platelets are fundamental vascular cells that regulate a myriad of physiologic processes through activation and interactions with immune and vascular cells. Although data are still emerging, accumulating evidence from clinical, translational, and basic science studies suggests that the process of aging results in alterations in redox biology and platelet function (summarized in Figure 2), which can have a significant impact on the development of vascular diseases. The regulation of these processes is complex and is impacted by a variety of changes that occur with aging, including inflammation, cellular stress, organelle dysfunction, and cell-to-cell interactions. Understanding the cellular and molecular redox mechanisms that drive these changes in platelets during aging will enhance our knowledge and will allow for the development of more targeted therapeutics.

## Figures and Tables

**Figure 1 antioxidants-11-00995-f001:**
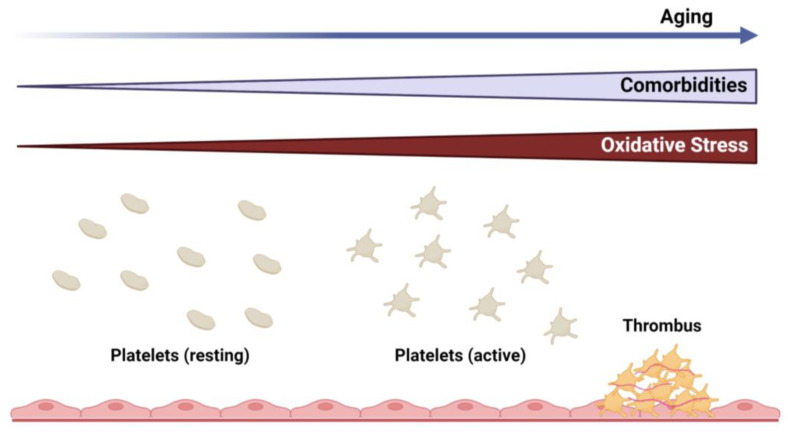
Overview of the relationship between aging and vascular disease, highlighting the pathogenic role of platelets. Cardiovascular comorbidities and oxidative stress are increased with advancing age, which can promote platelet activation and thrombus formation.

**Figure 2 antioxidants-11-00995-f002:**
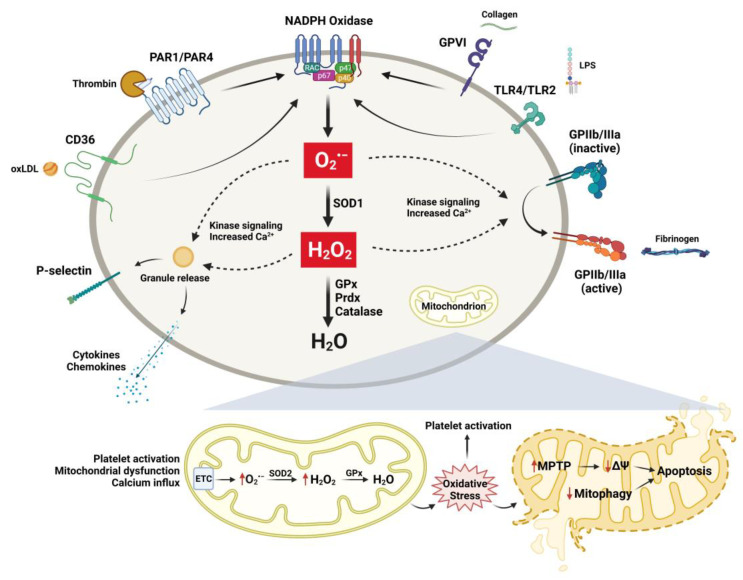
Selected mechanisms of platelet activation and ROS generation in aging and age-associated disease. A variety of extracellular ligands can associate with their respective platelet receptors and induce platelet activation and ROS generation. Multiple sources, including NADPH oxidases and mitochondria, can increase intracellular ROS in platelets. ROS generation in the cytoplasm can activate platelet surface receptors and induce granule release mediated by kinase signaling cascades and increased intracellular calcium. Dysfunctional mitochondria can produce increased ROS and result in mitochondrial permeability transition pore (MPTP) formation, loss of mitochondrial membrane potential (ΔΨm), and induction of platelet apoptosis. Antioxidant enzymes localized in both the cytoplasm and the mitochondria can reduce ROS and prevent platelet activation and apoptosis.

## Data Availability

Not applicable.

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
