# Peer review of "Redox Mechanisms of Platelet Activation in Aging"

_antioxidants, 2022, doi:10.3390/antiox11050995_

Round 1

Reviewer 1 Report

The authors have submitted a review in which provide a comprehensive overview about the redox mechanisms of platelet activation in aging.

It presents the most important findings regarding the current understanding of mechanistic links between oxidative stress and thrombotic vascular disease and the changes that occur with aging. The review is written by expert researchers in the field of oxidative stress and vascular diseases as shown by their numerous published articles.

In my opinion, the review is very well written, organized, very informative and timely, supported by 1 representative figure showing the selected mechanisms of platelet activation and ROS generation in aging; the bibliography has also a wide range of sources. 

I only have few minor concerns as follows:

Pag.2 line 117 and pag.3 line 142 the citations are not in the correct format.

I also recommend to create a paragraph apart for the COVID section

Author Response

RESPONSE TO REVIEWER 1

We would like to thank the Reviewers for their insightful evaluation of our review. The useful comments have served to strengthen the writing. The Reviewers have astutely identified weaknesses that need clarification or reorganization. We present a point-by-point response.

Peer-Reviewer 1:

 The authors have submitted a review in which provide a comprehensive overview about the redox mechanisms of platelet activation in aging. It presents the most important findings regarding the current understanding of mechanistic links between oxidative stress and thrombotic vascular disease and the changes that occur with aging. The review is written by expert researchers in the field of oxidative stress and vascular diseases as shown by their numerous published articles. In my opinion, the review is very well written, organized, very informative and timely, supported by 1 representative figure showing the selected mechanisms of platelet activation and ROS generation in aging; the bibliography has also a wide range of sources. 

We thank the Reviewer for the positive comments.

Minor comments:

Pag.2 line 117 and pag.3 line 142 the citations are not in the correct format.

Response:  We have corrected the format of the citations.

I also recommend to create a paragraph apart for the COVID section

Response: We thank the reviewer for the suggestion and have created a separate section titled “Platelets in COVID-19 Pathogenesis”.

RESPONSE TO REVIEWER 1

We would like to thank the Reviewers for their insightful evaluation of our review. The useful comments have served to strengthen the writing. The Reviewers have astutely identified weaknesses that need clarification or reorganization. We present a point-by-point response.

Peer-Reviewer 1:

 The authors have submitted a review in which provide a comprehensive overview about the redox mechanisms of platelet activation in aging. It presents the most important findings regarding the current understanding of mechanistic links between oxidative stress and thrombotic vascular disease and the changes that occur with aging. The review is written by expert researchers in the field of oxidative stress and vascular diseases as shown by their numerous published articles. In my opinion, the review is very well written, organized, very informative and timely, supported by 1 representative figure showing the selected mechanisms of platelet activation and ROS generation in aging; the bibliography has also a wide range of sources. 

We thank the Reviewer for the positive comments.

Minor comments:

Pag.2 line 117 and pag.3 line 142 the citations are not in the correct format.

Response:  We have corrected the format of the citations.

I also recommend to create a paragraph apart for the COVID section

Response: We thank the reviewer for the suggestion and have created a separate section titled “Platelets in COVID-19 Pathogenesis”.

Reviewer 2 Report

The authors present a review of platelets, reactive oxygen species and aging.  It is a worthwhile article for publication but needs significant re-writing in this reviewer's opinion.  It should not be difficult to appease me but it will take some work.  I object to the intermingling of human observations and murine research.  This leads to much confusion and is bext exemplified in paragraph lines: 205-227 where the authors discuss humans, then mice, and back to humans.  This manuscript will be much improved and needs to be re-written with subheadings in each section addressing human observations followed by subheading for murine models.  State what is known of human observations and then in a well defined section of animal experiments/results to explain what is likely in humans.  Do not describe human observation with animal models in the same paragraph.

I did not find syntax errors other than lines 59 and 62: flip the acronym ROS to line 62 and include reactive oxygen species (ROS) to line 59.

Author Response

We would like to thank the Reviewers for their insightful evaluation of our review. The useful comments have served to strengthen the writing. The Reviewers have astutely identified weaknesses that need clarification or reorganization. We present a point-by-point response.

 Peer-Reviewer 2:

 The authors present a review of platelets, reactive oxygen species and aging.  It is a worthwhile article for publication but needs significant re-writing in this reviewer's opinion.  It should not be difficult to appease me but it will take some work.  I object to the intermingling of human observations and murine research.  This leads to much confusion and is best exemplified in paragraph lines: 205-227 where the authors discuss humans, then mice, and back to humans.  This manuscript will be much improved and needs to be re-written with subheadings in each section addressing human observations followed by subheading for murine models.  State what is known of human observations and then in a well defined section of animal experiments/results to explain what is likely in humans.  Do not describe human observation with animal models in the same paragraph.

Response: We thank the reviewer for this helpful advice in improving the organization of the manuscript. We agree that in some instances, discussing different human and mice studies together may cause confusion, especially when the results are coming from different studies/manuscripts. We have reorganized the manuscript so that studies mainly focusing on observations in humans are typically presented in the beginning of the section/paragraph followed by studies mainly focusing on mice.

In some instances, both mice and human experiments were performed in the same study/manuscript, and since many of these results are directly complementary to each other for a set of endpoints/interpretations, we have decided to present these together to avoid disruption of the flow of the manuscript but made certain to explicitly state the type of organism each piece of data originated from.

For many of the sections, most of the results are from murine models with only a few pieces or no data from humans and vice versa; thus, it may not be relevant to separate into different subheadings of Human observations and murine models for each of the sections.

We hope that the revisions made have adequately addressed the concerns and improved the organization while maintaining the flow of the manuscript.

Minor comments:

I did not find syntax errors other than lines 59 and 62: flip the acronym ROS to line 62 and include reactive oxygen species (ROS) to line 59.

Response: We have corrected the location of the acronym “ROS” so that it appears with the first mention of “reactive oxygen species”.

Reviewer 3 Report

The MS is well organized and comprehensively described the role of oxidative stress in thrombotic vascular disease and the changes that occur with aging. 

Author Response

Thanks for your review!

Reviewer 4 Report

This is an interesting, well-written and somewhat comprehensive review of an important topic, namely the association between increased disease burden in aging with redox mechanisms, in the context of the blood cells platelets. We only have the following minor comments/suggestions:

  1. While not necessarily an aging issue, but smoking of various types is an important related issue that is worth adding to the present review article. The authors should consider adding a short paragraph or section, if they agree it will improve the manuscript.
  2. There are several typos that should be corrected (e.g., page 2, lines 59 and 62, the term reactive oxygen species should appear before the abbreviation "ROS" first appears; page 3, line 142 the reference # [44] is superscripted, etc.
  3. Figure 1 does not appear to be cited anywhere in the manuscript text.

Author Response

We would like to thank the Reviewers for their insightful evaluation of our review. The useful comments have served to strengthen the writing. The Reviewers have astutely identified weaknesses that need clarification or reorganization. We present a point-by-point response.

Peer-Reviewer 3

 This is an interesting, well-written and somewhat comprehensive review of an important topic, namely the association between increased disease burden in aging with redox mechanisms, in the context of the blood cells platelets. We only have the following minor comments/suggestions:

  1. While not necessarily an aging issue, but smoking of various types is an important related issue that is worth adding to the present review article. The authors should consider adding a short paragraph or section, if they agree it will improve the manuscript.

 Response: We have added a paragraph discussing the effects of smoking on platelet function based on the available literature (pg. 7 lines; 374-387).

 There are several typos that should be corrected (e.g., page 2, lines 59 and 62, the term reactive oxygen species should appear before the abbreviation "ROS" first appears; page 3, line 142 the reference # [44] is superscripted, etc.

Response: We have corrected the location of the acronym “ROS” so that it appears with the first mention of “reactive oxygen species”. We have corrected the format of reference 44.

  1. Figure 1 does not appear to be cited anywhere in the manuscript text.

Response: We have now cited Figure 2 (previously Figure 1) on page 8 in the conclusion paragraph, functioning as a summary figure for the review. We also added an additional figure (Figure 1) as recommended by the editor.

Round 2

Reviewer 2 Report

The authors have made some minor changes to appease me regarding structure, specifically to utilize subheadings to identify clearly content related to human observations and murine models.  I am not satisfied as there are still paragraphs containing both human and murine data that the authors rationalize.

Line 116 change mouse to "murine".

I am confused by paragraph lines 97-109 and lines 109-121.  Is the first portion all animal studies until line 106-109 and only the sentence in line 106-109 the only human observation?  I do not like the construction of the manuscript intermixing human and animal data!

In general, the manuscript is interesting and well-written.  Paragraph 122-133 is all human observations.  You could have just cut line 106-109 and placed it in the next paragraph with a subtitle of human observations.  I am not going to comment any further as the changes I requested to improve the manuscript were not made.

Author Response

RESPONSE TO REVIEWERS

Peer-Reviewer 2:

The authors have made some minor changes to appease me regarding structure, specifically to utilize subheadings to identify clearly content related to human observations and murine models.  I am not satisfied as there are still paragraphs containing both human and murine data that the authors rationalize.

I am confused by paragraph lines 97-109 and lines 109-121.  Is the first portion all animal studies until line 106-109 and only the sentence in line 106-109 the only human observation?  I do not like the construction of the manuscript intermixing human and animal data!

In general, the manuscript is interesting and well-written.  Paragraph 122-133 is all human observations.  You could have just cut line 106-109 and placed it in the next paragraph with a subtitle of human observations.  I am not going to comment any further as the changes I requested to improve the manuscript were not made.

Response: We thank the reviewer for the additional comments. We have moved lines 106-109 (discussing GPx and H2O2 in humans) from the murine data to the beginning of next paragraph. Due to this shift, the subsequent line numbers have changed from previous submission.  

We agree that there are a few other places where we discuss findings from human data in a paragraph that primarily has murine data (now lines 116-118, discussing reference #30) or vice versa (lines 136-138, reference #41). In these cases, the data are from the same study and are directly complementary to each other while supporting a single conclusion, so it’s not easy to discuss data from humans and murine models under a separate heading in these instances. So, to keep the flow uniform, we have not added separate subheadings for human and murine models. Similar types of data are discussed in other parts of the review (see below), where the findings in mouse and humans are complimentary and originated from same study:

Lines 212-222 discussing reference #82 that uses human and murine models to show mechanisms of platelet mitophagy

Lines 224-236 discussing reference #83 that uses both human and murine platelets to show a novel redox mechanism of mitophagy regulation by methionine oxidation in platelets.

Lines 247-270 discussing reference #91 that discusses elevated TNF-alpha in both aged humans and mice and effects on platelet activation.

We hope for the purposes of a comprehensive review (vs a results section or figure in a primary research article) some of the data from human and murine studies supporting a single conclusion may be discussed together. Taking suggestion from the reviewer, whenever appropriate, we have explicitly stated the host organism the data was derived from.

Line 116 change mouse to "murine".

We have made this correction (now line 117).